# A Single, Multimodal Exercise Tolerance Test Can Assess Combat Readiness in Army-ROTC Cadets: A Brief Report

**DOI:** 10.3390/jfmk8040152

**Published:** 2023-11-01

**Authors:** Derek A. Crawford, Katie M. Heinrich, Christopher K. Haddock, W. S. Carlos Poston, R. Sue Day, Christopher Kaipust, Blake Skola, Amanda J. Wakeman, Eric Kunkel, Addison Bell, Emily Wilhite, Nathanial Young, Allison Whitley, Madelyn Fritts

**Affiliations:** 1Department of Nutrition, Kinesiology, and Health, University of Central Missouri, Warrensburg, MO 64093, USA; skola@ndri-usa.org (B.S.); wakeman@ucmo.edu (A.J.W.); epk64130@ucmo.edu (E.K.); amb50060@ucmo.edu (A.B.); eew68580@ucmo.edu (E.W.); njy58580@ucmo.edu (N.Y.); anw89880@ucmo.edu (A.W.); mbf51100@ucmo.edu (M.F.); 2NDRI-USA, New York, NY 10001, USA; keithhaddock@hopehri.com (C.K.H.); carlosposton@hopehri.com (W.S.C.P.); day.susie@gmail.com (R.S.D.); kaipust@ndri-usa.org (C.K.); 3Department of Kinesiology, Kansas State University, Manhattan, KS 66506, USA

**Keywords:** military, performance, testing

## Abstract

The Army Combat Fitness Test (ACFT) is a multi-event assessment battery designed to determine the combat readiness of U.S. Army personnel. However, for Reserve Officers’ Training Corps (ROTC) programs the logistical demands of collegiate life make repeated administration of the ACFT challenging. The present study sought to design and evaluate a single, multimodal exercise tolerance test (METT) capable of serving as a time-efficient proxy measure of combat readiness. Methods: Using a formal instrument design process, we constructed the METT to mimic the demands of the ACFT and assessed its reliability, validity, and responsiveness. Results: The METT demonstrates minimal measurement error (i.e., a 2% coefficient of variation), concurrent validity with the ACFT (R^2^ = 0.327, F = 10.67, *p* < 0.001), the ability to classify cadets who may be at-risk for failing the ACFT (X^2^ = 8.16, *p* = 0.017, sensitivity = 0.878, specificity = 0.667), and appropriate change following a training intervention (5.69 ± 8.9%). Conclusions: The METT has the potential to provide a means to monitor progress, identify areas for improvement, and guide informed decision-making regarding individualization of cadet combat training plans.

## 1. Introduction

The U.S. Army Reserve Officers’ Training Corps (ROTC) is the primary preparatory program for college students who wish to pursue a career in the Army [1]. Throughout the program, students are expected to meet specific body composition recommendations and physical fitness standards [1]. With respect to physical fitness, cadets complete the Army Combat Fitness Test (ACFT) as the assessment of record used to determine their combat readiness [2]. The ACFT is composed of six individual physical tests (i.e., a 3-repetition maximum deadlift, a sprint-drag-carry task, hand release push-ups, a plank task, a standing medicine ball throw, and a 2-mile run time to completion) [2]. The ACFT individual component tasks are performed within the same session, with adequate rest in between, resulting in a test battery lasting approximately 70 to 120 min in duration [2]. This time demand, coupled with large cadres (e.g., 100+ cadets) with often differing schedule demands and equipment constraints, presents a logistical challenge when administering the ACFT in collegiate settings.

Fortunately, Army Command only requires the administration of the ACFT once per academic year for ROTC programs. However, while this alleviates logistical challenges, cadets are ultimately judged and potentially assigned supplemental physical training based on their performance during the ACFT. With limited availability to test and re-test cadets using the ACFT, there is a need for a simpler assessment method that may provide more flexibility to those designing and implementing ROTC physical training programs. A time-efficient, comprehensive assessment of cadets’ progress during training would allow for more nuanced decisions on exercise programming variables to improve combat readiness and whether additional training is necessary.

Recently, physical work capacity is posited as a unique metric of human fitness that is distinct from traditional measures such as VO_2MAX_, muscular strength, etc. [3]. Assessment of work capacity necessitates temporally combing multimodal (e.g., many different bio-motor abilities) physical tasks and completing them at a relatively high intensity (i.e., as fast as possible) [4]. Additionally, work capacity assessments can be designed in a way as to be “task-specific”, in that, the elements included within the assessment and the time domain assessed determine the utility for predicting performance outcomes of interest. For example, in an athlete population, a work capacity assessment attempting to predict on-field performance would need to include elements and be of a duration that simulate technical skill and bioenergetic demands. With the bio-motor abilities associated with combat readiness already defined (i.e., the ACFT component tasks), it should be feasible to develop a work capacity assessment to predict combat readiness. 

Thus, the purpose of the present study was to develop and preliminarily evaluate a multimodal exercise tolerance test (METT) of physical work capacity as it relates to predicting combat readiness using the ACFT as a criterion measure. Such a test would provide ROTC leadership with the ability to effectively monitor cadet progress more efficiently during an academic year. This may potentially result in lower recidivism rates and higher rates of advancement to active service among cadets via better designed and/or tailored physical training programs. We expect to develop a METT capable of predicting, and discriminating, ACFT performance. 

## 2. Materials and Methods

### 2.1. Participants

Table 1 provides the sex-specific biometric data of the 47 cadets (28% female) who agreed to participate in this study. Participants were required to be an active member of the University’s Army ROTC program for at least one semester prior to study commencement. All participants reported no significant disease or health conditions (e.g., diabetes) that may have contraindicated participation in vigorous exercise testing and/or programs. Written informed consent was obtained prior to study participation and all procedures were approved by the University Institutional Review Board for Protection of Human Research Subjects. 

### 2.2. Development of the Multimodal Exercise Tolerance Test (METT) 

Over the course of two academic semesters, the authors employed the Instrument Development Process to conceptualize, design, and beta-test a METT to assess work capacity related to combat readiness [5]. This process consisted of (1) concept identification, (2) test construction, (3) reliability, and (4) validity testing. Within the fall academic semester, the author group focused on concept identification (i.e., identifying physical capacities of combat readiness that need to be challenged and underlying bioenergetic demands) and test construction (i.e., exercise/movement task selection, intensity/volume, and duration needs) phases. Using the ACFT components as a framework, the following physical capacities were identified as contributing to combat readiness: *lower and upper extremity strength/power*, *total body stability/coordination*, *lower and upper extremity muscular endurance*, *maximal running speed*, *change of direction ability*, and *cardiovascular endurance*. Figure 1 outlines the final construction of the METT for assessing combat readiness. 

Once the test was constructed, an iterative stimulus (i.e., exercise loads and volume) and design (e.g., logistical constraints of test administration) optimization phase took place. Within this phase, loading intensities for all externally loaded exercises were optimized for maximizing the amount of work (in ft-lbs measured using an optical linear transducer; EliteForm Integrated, EliteForm, Lincoln, NE, USA) that a college-aged individual could complete in the anticipated time domain of the task. For example, for the targeted repetitions for the deadlift task (i.e., 9) were able to be complete in an average of 15 s. Loads (as a percentage of body mass) were systematically increased until work performed in 15 s peaked. 

Figure 2 illustrates the recommended setup for administration of the METT. To maximize the ecological validity of the assessment, the distance constraints for the sprint/agility circuit and farmer’s carry tasks were set as to allow for two testing “lanes” to fit within a standard size basketball court. This allowed for ROTC administrators to logistically assess multiple cadets at a time within their regular training sessions and facilities. 

### 2.3. Physical Testing Procedures 

#### 2.3.1. Body Measurements

During a laboratory visit, height was measured to the nearest 0.1 cm using a wall-mounted stadiometer, weight was measured to the nearest 0.1 kg using a medical-grade scale (PS-7700, Befour, Inc., Saukville, WI, USA), and body composition was measured by dual X-ray absorptiometry (DXA; Prodigy iDXA, General Electric, Boston, MA, USA). Cadets were instructed to refrain from exercise and alcohol consumption for a 12-h period immediately preceding their body composition assessment. 

#### 2.3.2. Combat Readiness

All participants completed the ACFT in accordance with the procedures outlined in the Army fitness manual [2]. Consisting of the six events noted in the introduction (e.g., hand release pushups, medicine ball throw, etc.), each event is associated with a 100-point score with a maximum of 600 points possible. The event scores were standardized into z-scores and a total score of performance (ACFT_TSP_) was created by calculating the mean of each cadet’s z-scores to represent a single measure of combat readiness relative to the cadre. In addition to this continuous variable, a dichotomous variable was created to identify those cadets who either “passed” or who “failed”/were “at-risk” of failing the ACFT following the criteria of scoring below 60 points in any one event or in the 60 s across more than two events, respectively. 

#### 2.3.3. METT Procedures

Cadets complete the METT as part of a regularly scheduled training session during the 2nd and 15th week of the academic semester. During the assessment, cadets were observed by a peer who used a stopwatch to record their time to completion for each individual task to the nearest 0.1 s. All task times were added together to reflect total test time. Cadets were provided demonstration of movement standards and allowed to practice movements prior to testing. Cadets completed the METT as fast as possible. Data collection sheets were returned to research staff who checked data accuracy and entered them into a spreadsheet for later analyses. 

### 2.4. Experimental Procedures 

All baseline assessments and physical testing procedures were conducted during the first 2 weeks of the Spring 2023 academic semester. Cadets then completed an 11-week High Intensity Functional Training (HIFT) intervention designed to increase general work capacity [6]. The general framework for the periodization and progression of the HIFT intervention were developed by the research team, but specific details regarding exercise selection, scheduling, and delivery location were made by cadre training staff members. Following the completion of an 11-week HIFT intervention, body measurements, the ACFT, and METT tests were completed again over a 2-week period. Only baseline data were used to assess the validity and reliability of the METT while baseline and post-intervention data were used to determine the responsiveness of the test. 

### 2.5. Data Analyses

Descriptive statistics were calculated for all variables across both the baseline and post-intervention. Normality was assessed using the Shapiro–Wilks test. All analyses were performed using the R statistical programming language (v. 4.1) within the Jamovi graphical user interface (v. 2.3) [7,8]. A Pearson’s correlation coefficient matrix for all variables was calculated to assess content validity of METT components. The *GAMLj: General analysis for linear models* package was used to construct a model to assess the concurrent validity of the METT for predicting combat readiness [9]. Discriminant validity was assessed using a logistic regression model and the associated cut-off scores for passing or failing the ACFT were estimated using a Receiver Operator Characteristic curve via the *ROCR: Visualizing the performance of scoring classifiers* package [10]. Change scores (% change) for METT time to completion, and descriptive statistics, were calculated to evaluate the measure’s responsiveness. An alpha level of 0.05 was used for all inferential analyses. 

## 3. Results

### 3.1. METT Reliability 

During the development process, beta testing procedures within the research team provided estimates of the mean within-subjects (i.e., test-retest) and mean assessment (i.e., intra- and inter-rater) variation in time to completion (in seconds) for the METT. Using these pilot data, within-subjects’ coefficient of variation (CV) was observed to be 1.3% of the time to completion. The intra-rater CV was observed to be 0.02% and inter-rater CV was observed to be 0.73% of the time to completion. The authors recommend measurement error intervals for the METT be estimated as the sum of the potential within-subject (i.e., 1.3%) and inter-rater (i.e., 0.73%) CV, equaling an estimated ± 2.03% measurement error. 

### 3.2. METT Validity 

#### 3.2.1. Content Validity

Table 2 presents the correlation matrix between the individual components of the METT (time to completion in seconds) and ACFT assessments (percentile rank). Each individual component of the METT demonstrates a significant, negative correlation with at least one component of the ACFT. Of note, some discrepancies exist between METT components and their hypothesized ACFT counterparts. For example, METT Task 2 (i.e., deadlift repetitions) does not correlate with the three-repetition maximum deadlift load within the ACFT (*p* = 0.996) and METT Task 1 (i.e., sprint/agility drill) does not correlate with the sprint-drag-carry task within the ACFT (*p* = 0.109) as was expected. However, other components of the METT align with their targeted ACFT counterparts as intended. For example, total time to completion and METT Task 4 (i.e., farmer’s carry) significantly correlated with the 2-mile run performance and the plank task within the ACFT, respectively. Interestingly, the farmer’s carry (METT Task 4) best related to ACFT tasks, being significantly correlated with five out of six components (with the three-repetition maximum deadlift as the outlier). 

#### 3.2.2. Concurrent Validity

Table 3 provides the parameter estimates for a general linear model for predicting combat readiness (i.e., the composite ACFT_TSP_ variable) using the METT time to completion (scaled to sample z-scores) at the baseline. As expected, there is a significant, negative relationship wherein a 1-standard deviation decrease in the METT time to completion results in a 0.460-standard deviation increase in the AFCT performance. 

Figure 3 presents a scatterplot of the raw data split by sex. The trendlines for sex-specific data are included. While the effect of sex within the general linear model presented in Table 3 is not significant, there are large effects in the METT time to completion (difference = 32.2 ± 16.6 s, t = 1.95, *p* = 0.054, Cohen’s *d* = 0.387) between males (mean = 430 ± 83.8 s) and females (mean = 499 ± 64.7 s). Visually, between sexes, there also appears to be a potential difference in the relationship between the baseline METT performance and combat readiness. For these reasons, the authors determined that sex, as a control variable, would be carried forward into the analyses evaluating the discriminant validity of the METT.

#### 3.2.3. Construct (Discriminant) Validity

Table 4 presents the parameter estimates for a logistic regression model for predicting ACFT “At-risk/Fail” from the METT time to completion, controlling for sex. Within the model, the METT time to completion significantly predicts the probability of being at-risk or failing the ACFT, wherein there is a 1.6% increased risk of failure for every 1-s increase in total test time above the cut-off score(s). Panel A in Figure 4 illustrates both the probability distributions for sensitivity and specificity measures of potential cut-off values. Panel B in Figure 4 illustrates the receiver operator characteristic (ROC) curve for the cut-off probability of 0.95. Using the derived parameter estimates from Table 4, the calculated thresholds for METT time to completion (accounting for measurement error) required to “pass” the ACFT are less than 536 ± 11 s and 572 ± 12 s for males and females, respectively. Additionally, the cutoff scores demonstrate a reasonable degree of diagnostic utility (e.g., sensitivity, specificity, etc.) as evidenced in Table 4.

### 3.3. METT Responsiveness

Following the 11-week HIFT intervention (designed to improve the work capacity), the mean change in the METT time to completion was −5.69 ± 8.9% (95%CI: −8.33, −3.05%). Figure 5 displays the individual-level change in the METT time to completion across the baseline time to completion z-score distribution. Across the study sample, 32 out of 47 (65.3%) cadets improved their METT performance beyond the measurement error, with a maximum improvement of approximately 27.5%. Conversely, 9 out of 47 (19.1%) cadets regressed their METT performance beyond the measurement error, with a maximum regression of approximately 25%. These proportions of individual response heterogeneity align with published data from large-scale exercise interventions [11]. Additionally, there is no significant relationship (F_1,46_ = 0.885, *p* = 0.352, R^2^ = 0.019) between change scores and the baseline METT performance, indicating no bias in responsivity due to the baseline METT performance status.

## 4. Discussion

This study sought to develop and preliminarily evaluate a METT of physical work capacity capable of predicting combat readiness in a collegiate Army ROTC cadet. By employing the Instrument Development Process [5], we designed a time-efficient, single test capable of delivering a valid, reliable, and responsive proxy measure of combat readiness.

At the time of writing, we are aware of only one other attempt to develop a single test capable of serving as a proxy measure of combat readiness. Moore et al. (2022) designed and assessed the reliability of a test named the Combat Readiness Assessment (CRA) [12]. The CRA consists of seven physical task “zones” spread across the field space within a 400-m track and field complex. These zones are made up, in order, of a 10-m sprint zone, 30-m loaded carry zone, 20-m weight drag zone, another 10-m sprint zone, a 15-m “agility box” zone, and a final 60-m sprint zone. While the authors should be commended for their efforts in conceptualizing the CRA, there are ultimately limitations in its construction and assessment that restrict its utility for serving as a proxy measure of combat readiness.

First, compared to the METT designed in the present study, the CRA does not appear to assess the domains of *lower and upper extremity strength/power* or *cardiovascular endurance*. While the CRA does include a loaded carry task, unfortunately the loading scheme is not noted, making determination of its ability to challenge strength and/or power difficult. The METT includes both lower (i.e., deadlift) and upper extremity (i.e., push press) exercise tasks specifically optimized at loading schemes designed to maximize physical work. With respect to cardiovascular endurance, the mean time to completion of the CRA was 208 ± 21 s. Conversely, the mean time to completion of the METT was 464 ± 73 s across both sexes. While it takes approximately 3 min to achieve steady state oxygen consumption, tests of cardiovascular endurance typically take much longer (i.e., 8–12 min) [13]. Maximal tests like critical power typically last no longer than 3 min, yet these tests are designed to capture an individual anerobic rather than aerobic capacity [14]. Thus, when a cadet is instructed to give maximal effort from the onset of a test it can be reasonably assumed that a longer duration effort will better represent cardiovascular endurance/capacity. Such is the case comparing the CRA to the METT.

Second, in evaluating the CRA, the authors only choose to investigate the test’s reliability. Conversely, herein we report the reliability, validity, and responsiveness of the METT. To compound this limitation, Moore et al. (2022) also note that the findings regarding the short-term reliability of the CRA are inconclusive given that the test time to completion yielded relatively high coefficients of variation (i.e., 10%) across familiarization sessions. In contrast, the coefficients of variation observed in beta testing of the METT were much smaller for test-retest (i.e., 1.3%), intra-rater (i.e., 0.02%), and inter-rater (i.e., 0.73%) reliability. One reason for this discrepancy in reliability may be found in the tests’ duration. With the METT being nearly twice as long as the CRA, it may be that its time to completion is less prone to acute changes in motivation, hydration, sleep, etc., leading up to the testing session.

The results of this study have provided a valid, reliable, and ecologically sensitive test that can serve as a proxy measure of combat readiness in lieu of having cadets complete the lengthy ACFT. The METT demonstrates a reasonable degree of measurement error (i.e., 2.03%) across multiple repeated assessments at the same relative timepoint (i.e., back-to-back weeks) and between observers administering the test. Furthermore, the METT exhibited good content validity as there were significant moderate correlations between the individual components of the test and those of the ACFT. It also demonstrated concurrent validity as baseline scores on the METT were able to significantly predict performance on the ACFT at the baseline. As a diagnostic tool, the METT was able to discriminate between cadets who were at-risk for, or failed, the ACFT from those who were not with a high degree of sensitivity (100%). We also provide sex-specific cut-off scores for time to completion (i.e., males = 536 ± 11 s, females = 572 ± 12 s) on the METT that, above which, ROTC administrators can be confident that cadets would likely not pass the ACFT.

Practically, our recommendation is that this newly developed METT be used as a monitoring tool within ROTC programs to make more informed decisions with respect to cadets’ individual training plans. For example, due to the ease of implementation within existing training schedules, the METT could be conducted monthly to determine overall cadre progress toward a target training objective such as “x percent of cadets are predicted to pass the ACFT”. Alternatively, these monthly assessments could also determine if individual cadets need additional training beyond what is currently planned if they are classified as at-risk of failing the ACFT. With some minor changes in the test administration (i.e., recording of time to completion for each task), it would even be possible to use the METT as a prescriptive tool by identifying relative deficiencies and highlighting them within training programming. The time-efficient nature of the METT would indeed provide a degree of flexibility when looking to adjust training programs at the individual level.

The present study has several key considerations. Chief among the relative strengths is the comprehensive evaluation of METT reliability, validity, and responsiveness. Another strength is that the practicality of our assessment of discriminant validity provides actionable insights on using the METT to classify cadets based on their predicted performance on the ACFT. Two limitations of this study should be noted. First, the sex distribution in this study is skewed in favor of male cadets. Therefore, there is limited ability to generalize the findings of this study to females. However, even though we chose to carry forward sex as a predictor variable in the logistic regression analyses of discriminant validity, it should be noted that sex was a non-significant factor within the original general linear model. Second, the reliability analyses conducted within the present study used pilot data from the research team rather than from the cadet sample. It is plausible that measurement error within this group is different from that observed in our research staff. Future research should look to correct these limitations by repeating the present study using a larger, more balanced cadre that includes more formalized reliability evaluation such as in the study assessing the CRA. Such a study will have the potential to further refine the cut-off scores presented in this study for determining predicted ACFT performance.

## 5. Conclusions

This study developed and evaluated the METT for predicting combat readiness within a collegiate Army ROTC cadre. Using the Instrument Development Process, we constructed a time-efficient, singular test that provides a valid, reliable, and responsive proxy measure of ACFT performance. The METT has the potential to provide a means to monitor progress, identify areas for improvement, and guide informed decision-making regarding individualization of cadet training plans. While acknowledging the current study’s limitations in sex distribution and reliability analyses, this preliminary evaluation of the METT can guide further investigation and refinement of this tool.

## Figures and Tables

**Figure 1 jfmk-08-00152-f001:**
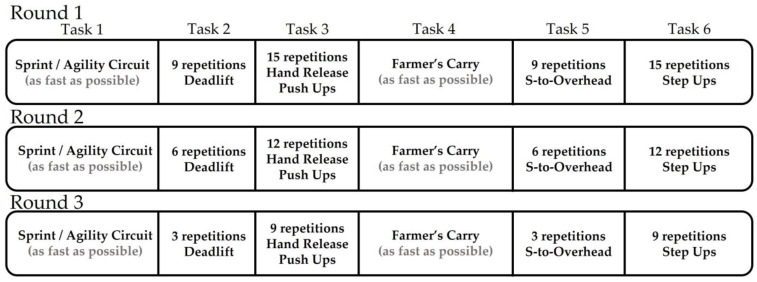
Outline of Components within the METT for ROTC Cadets. For the spring/agility circuit, cadets sprint forward for 94 feet, right shuffle 15 feet, backpedal 15 feet, left shuffle 15 feet, then reverse the path returning to the start position. For the deadlift task, both sexes use 75% of body mass loaded on a trap bar. This same load is used during the farmer’s carry task in which cadets carried the load 94 feet down and 94 feet back to the starting position. For the shoulder-to-overhead task, loads are 50% and 40% of body mass for males and females, respectively, loaded on a standard barbell. For the step-up task, implement heights are 16 and 20 inches for females and males, respectively.

**Figure 2 jfmk-08-00152-f002:**
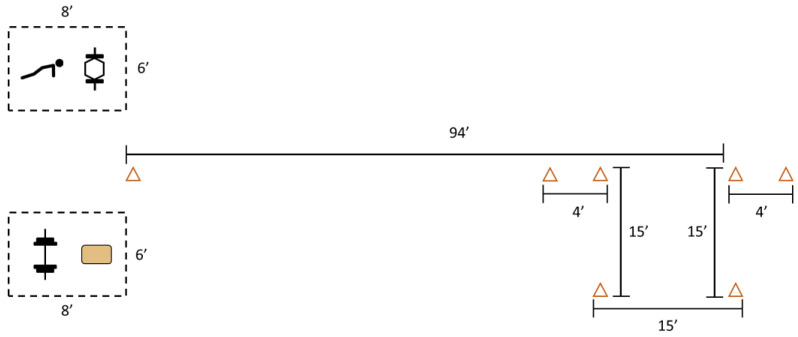
Suggested layout for the METT for ROTC cadets. Orange triangles represent cones, hexagon image represents the trap bar, tan box represents the step-up instrument, dashed lines are to only suggest space allocation(s) for the deadlift, push-up, shoulder-to-overhead, and step-up tasks. Dimensions are reported in feet.

**Figure 3 jfmk-08-00152-f003:**
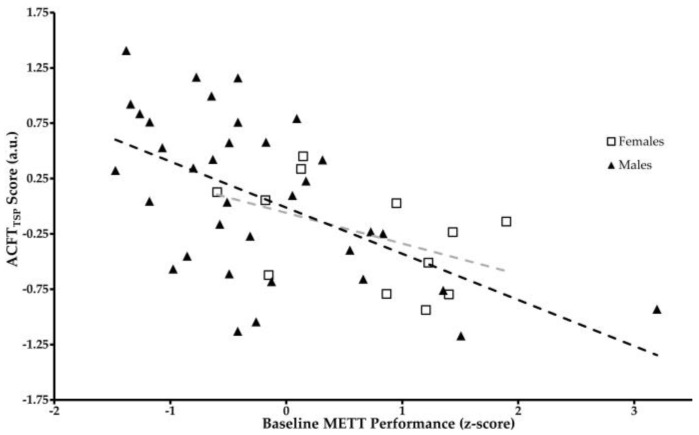
Relationship between baseline ACFT and METT performance. Darker trendline represents the relationship for males and lighter trendline represents the relationship for females.

**Figure 4 jfmk-08-00152-f004:**
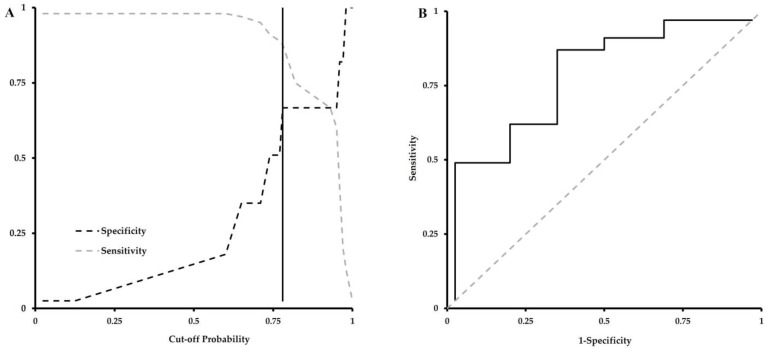
Sensitivity and Specificity Analyses Supporting Figures. Panel (**A**) illustrates the sensitivity and specificity values across the cut-off probability distribution. The vertical line highlights the cut-off value (0.95) which maximizes these metrics. Panel (**B**) is the associated ROC curve for the 0.95 value.

**Figure 5 jfmk-08-00152-f005:**
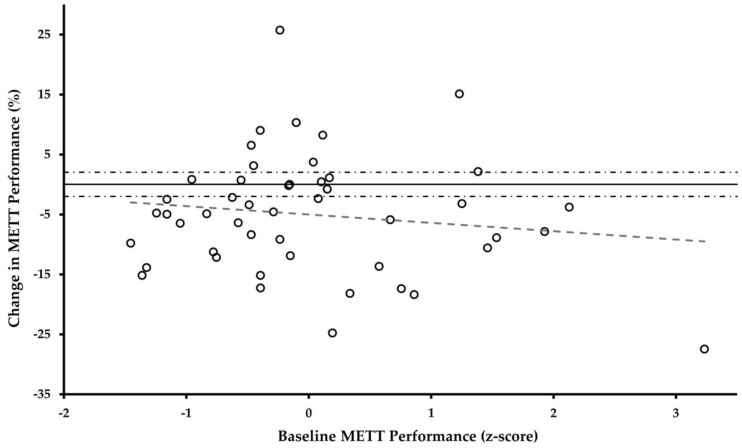
Relationship between Changes in METT Time to Completion and Baseline Performance. Solid line indicates change of 0%. Dashed and dotted lines represent the thresholds of measurement error (i.e., 1.3% within-subject variation + 0.73% inter-rater variation = 2.03% error).

**Table 1 jfmk-08-00152-t001:** Sex-specific biometric data.

	Female	Male
*n*	13	34
Age	20.6 ± 1.0	21.2 ± 2.1
Height (cm)	164.0 ± 6.8	179.0 ± 6.4
Weight (kg)	63.3 ± 9.2	82.8 ± 12.2
FFM (kg)	41.8 ± 4.5	59.1 ± 12.8
BF%	30.5 ± 4.9	23.1 ± 5.2
BMD (g/cm^2^)	1.23 ± 0.1	1.35 ± 0.1
Years in ROTC	2.46 ± 0.8	2.62 ± 1.2

**Table 2 jfmk-08-00152-t002:** Correlation matrix for METT and ACFT individual components (*n* = 47).

	M (SD)	3DL	SPT	HRP	PLK	SDC	2-MR
Sprint/Agility Time (s)	28.9 ± 5.2	−0.223	−0.087	−0.108	**−0.478 ***	−0.237	**−0.526 ***
Deadlift Time (s)	14.5 ± 5.0	0.001	0.048	−0.318 *	−0.401 *	0.029	−0.129
HRP Time (s)	27.3 ± 10.6	−0.288 *	−0.18	**−0.563 ***	−0.223	−0.286	−0.373 *
Farmer’s Carry Time (s)	34.6 ± 8.2	−0.242	−0.294 *	−0.331 *	**−0.518 ***	−0.338 *	**−0.598 ***
S-to-Overhead Time (s)	19.6 ± 11.3	−0.126	−0.075	**−0.498 ***	−0.371 *	−0.237	−0.284
Step Ups Time (s)	29.5 ± 10.5	−0.142	−0.049	−0.269	−0.164	−0.26	−0.338 *
Total Test Time (s)	447.9 ± 84.6	−0.253	−0.24	**−0.545 ***	**−0.473 ***	−0.331 *	**−0.481 ***

* indicates *p* < 0.05, bold indicates *p* < 0.001; M = mean, SD = standard deviation, 3DL = three-repetition maximum deadlift, SPT = standing power throw, HRP = hand release push-ups, PLK = plank, SDC = sprint-drag-carry, 2-MR = 2-mile run; S-to-Overhead = shoulder to overhead press.

**Table 3 jfmk-08-00152-t003:** General linear model output for predicting ACFT_TSP_ (*n* = 47).

Overall Model Test		R^2^ = 0.486	F_(2,44)_ = 13.54	*p* < 0.001
*Parameter*	**Coefficient**	**SE**	**95% CI**	***p*-value**
Total Time (s) *	−0.460	0.083	−0.63, −0.29	<0.001
Sex ^#^	0.589	0.240	0.10, 1.07	0.018
Weight (lbs)	0.012	0.003	0.00, 0.01	<0.001

* Total time is scaled to sample z-scores, ^#^ sex is dummy coded using male = 1 and female = 0.

**Table 4 jfmk-08-00152-t004:** Logistic regression model output and sensitivity analyses for predicting At-risk/Fail on ACFT (*n* = 47).

Overall Model Test			R^2^_McF_ = −0.227	X^2^_(2,44)_ = 8.16	*p* = 0.017
*Parameter*		**Coefficient**	**SE**	**95% CI**	***p*-value**
Intercept		9.8515	3.560	2.881, 15.897	0.006
Total Time (s)		−0.0166	0.007	−0.029, −0.003	0.021
Sex: Female-Male		0.6096	1.118	−1.581, 2.799	0.586
**2 × 2 Classification Table**				**Predictive Measures**	
	Predicted			Accuracy	0.553
Observed	At-risk/Fail	Pass	% Correct	Specificity	1.000
At-risk/Fail	6	0	1.00	Sensitivity	0.488
Pass	21	20	48.8	Area Under the Curve	0.817

## Data Availability

The de-identified data that support the findings of this study are available on request from the corresponding author. Any use of these data should be accompanied by appropriate attribution to this published report.

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
