# Peer review of "A Single, Multimodal Exercise Tolerance Test Can Assess Combat Readiness in Army-ROTC Cadets: A Brief Report"

_jfmk, 2023, doi:10.3390/jfmk8040152_

Round 1
Reviewer 1 Report
Comments and Suggestions for Authors
- Lines 109-111: The authors mention that the reps were for the “average cadet” to perform in 15 seconds. How was average cadet quantified? If this was based upon the university’s program, then the transferability is limited.
- The authors mention in the introduction that the ACFT is equipment intensive and that something like the METT would be beneficial do to less logistical challenges. However, by having the barbell for the shoulder press, this is additional equipment above what is needed for the ACFT, if programs just have the ACFT equipment, they could not do this METT.
- What was the time between the ACFT and the METT?
- Limited information of the sensitivity and specificity component is concerning. How was this calculated? (i.e., What was considered a true positive? Accurately predicant the ACFT failure or accurately predicting the ACFT pass?) There are also different results provided in the table than in the discussion. More information on this aspect is needed before a judgement on acceptability for publication can be made from this reviewer.
- The authors report that fallowing an 11-week training program METT increased but do they have data on if ACFT increased? If the METT is supposed to predict ACFT performance, did those who improve on the METT actually improve on ACFT?
- Line 262: Typically in ROTC the term cadre refers to the leadership of active duty soldiers not cadets. I suggest changing this to be cadets unless you truly are referring to cadre.
- Description of the cadets fitness level or at least ACFT demographics
- How was the time of the task defined? If they did 3 rounds of each task were all three times added together?
- How was the concept identification obtained? Was this a Delphi process? More information is needed on how the METT was developed. For example: how was consensus on the tasks included, the reps per task and the decision to do 3 rounds (still unclear) with decreased reps per round.
- The integration of available literature is limited. While there are understandable limitations to placing the current project into the large context of work capacity assessments relative to the ACFT, are there other tactical population work capacity tasks or even those in the sports environment that this project could be compared to in the discussion?
- Table 2 would be more interpretable if descriptions of each task were given rather than them being numbered and the reader having to go back to the figure to ID the task.
- From the table it appears that only 6 cadets were “at risk” or failed the ACFT. This is a small group of a subpopulation that the METT hopes to ID, but this was not listed as a limitation.
- Based upon the figure, the METT is performed in three rounds. Please clarify.
Author Response
Lines 109-111: The authors mention that the reps were for the “average cadet” to perform in 15 seconds. How was average cadet quantified? If this was based upon the university’s program, then the transferability is limited. Changed to state "a college-aged individual"
- The authors mention in the introduction that the ACFT is equipment intensive and that something like the METT would be beneficial do to less logistical challenges. However, by having the barbell for the shoulder press, this is additional equipment above what is needed for the ACFT, if programs just have the ACFT equipment, they could not do this METT. Noted in the discussion now.
- What was the time between the ACFT and the METT?
- Limited information of the sensitivity and specificity component is concerning. How was this calculated? (i.e., What was considered a true positive? Accurately predicant the ACFT failure or accurately predicting the ACFT pass?) There are also different results provided in the table than in the discussion. More information on this aspect is needed before a judgement on acceptability for publication can be made from this reviewer. It is clarrified in lines 228-229 that the reference condition is "at-risk/failing the ACFT" is what we want to predict.
- The authors report that fallowing an 11-week training program METT increased but do they have data on if ACFT increased? If the METT is supposed to predict ACFT performance, did those who improve on the METT actually improve on ACFT? We agree this would be fantastic information to include! Unfortunately, do to circumstances outside our control there is no post-intervention ACFT data avaliable.
- Line 262: Typically in ROTC the term cadre refers to the leadership of active duty soldiers not cadets. I suggest changing this to be cadets unless you truly are referring to cadre. Corrected.
- Description of the cadets fitness level or at least ACFT demographics
- How was the time of the task defined? If they did 3 rounds of each task were all three times added together? This is now clarified in lines 146-147.
- How was the concept identification obtained? Was this a Delphi process? More information is needed on how the METT was developed. For example: how was consensus on the tasks included, the reps per task and the decision to do 3 rounds (still unclear) with decreased reps per round. The developmenet process is already clearly stated within the section entiled "Development of the Multimodal Exercise Tolerance Test (METT)".
- The integration of available literature is limited. While there are understandable limitations to placing the current project into the large context of work capacity assessments relative to the ACFT, are there other tactical population work capacity tasks or even those in the sports environment that this project could be compared to in the discussion? Unfortunately, the conceptualization of work capacity (as we define it) has not been investigated in many contexts and/or populations. This study is one of the first to create a specific work capacity test for a specific purpose/ population.
- Table 2 would be more interpretable if descriptions of each task were given rather than them being numbered and the reader having to go back to the figure to ID the task. We agree. The change has been made.
- From the table it appears that only 6 cadets were “at risk” or failed the ACFT. This is a small group of a subpopulation that the METT hopes to ID, but this was not listed as a limitation. That figure is only specific to this sample. Large scale data suggest that around 28% of potential cadets fail the ACFT the first time.
- Based upon the figure, the METT is performed in three rounds. Please clarify. Yes, this is correct.
Reviewer 2 Report
Comments and Suggestions for Authors
The present study sought to design and evaluate to design and evaluate a single, multimodal exercise tolerance test of physical work capacity as it relates to predicting combat readiness using the Army Combat Fitness Test as a criterion measure.
The report is well articulated and well written. I congratulate the authors for their outstanding effort. Nothing to add from my side.
Author Response
We would like to thank the reviewer for their kind comments on the work. The manuscript has been revised to meet other reviewers concerns.
Reviewer 3 Report
Comments and Suggestions for Authors
Dear Authors
You have written an interesting manuscript focused on the design and evaluation of a single, multimodal exercise tolerance test (METT) capable of serving as a time-efficient proxy measure of combat readiness.
However, some parts need to be addressed for greater clarity.
Abstract: please report the number of study participants.
The introduction is well-written, clearly leads to the main study rationale and is well backed up by relevant and up-to-date references.
Methods: Please report how you determined the sample size and the sampling method. Did you use G*Power or any other method?
Table 1 - explain abbreviations in the table
Figure 1 - what was the break between the tasks - this is not reported anywhere.
Figure 2 could also have tasks marked for even better clarity
Body measurements - at what part of the day were they done? report
The statistical analysis is robust and well done.
Line 266 - what about https://pubmed.ncbi.nlm.nih.gov/31800475/
Comparison to CRA is well established.
The limitations are well stated.
Overall a well-written paper that needs minor amendments.
Kind regards
Comments on the Quality of English Language
Minor editing of the English language required.
Author Response
We would like to thank the reviewer for their comments. All of your comments/concerns were addressed within the responses to Reviewer #1 and are now reflected in the revised manuscript.
Round 2
Reviewer 1 Report
Comments and Suggestions for Authors
No additional comments.